Automated Knowledge Base Construction (2019)                    Conference paper

# Answering Visual-Relational Queries in Web-Extracted Knowledge Graphs

**Daniel Oñoro-Rubio**                          DANIEL.ONORO@NECLAB.EU
**Mathias Niepert**                             MATHIAS.NIEPERT@NECLAB.EU
**Alberto García-Durán**                        ALBERTO.DURAN@NECLAB.EU
**Roberto González-Sánchez**                    ROBERTO.GONZALEZ@NECLAB.EU
*NEC Labs Europe*

**Roberto J. López-Sastre**                     ROBERTOJ.LOPEZ@UAH.ES
*University of Alcalá*

## Abstract

A visual-relational knowledge graph (KG) is a multi-relational graph whose entities are associated with images. We explore novel machine learning approaches for answering visual-relational queries in web-extracted knowledge graphs. To this end, we have created IMAGEGRAPH, a KG with 1,330 relation types, 14,870 entities, and 829,931 images crawled from the web. With visual-relational KGs such as IMAGEGRAPH one can introduce novel probabilistic query types in which images are treated as first-class citizens. Both the prediction of relations between unseen images as well as multi-relational image retrieval can be expressed with specific families of visual-relational queries. We introduce novel combinations of convolutional networks and knowledge graph embedding methods to answer such queries. We also explore a zero-shot learning scenario where an image of an entirely new entity is linked with multiple relations to entities of an existing KG. The resulting multi-relational grounding of unseen entity images into a knowledge graph serves as a semantic entity representation. We conduct experiments to demonstrate that the proposed methods can answer these visual-relational queries efficiently and accurately.

## 1. Introduction

Numerous applications can be modeled with a knowledge graph representing entities with nodes, object attributes with node attributes, and relationships between entities by directed typed edges. For instance, a product recommendation system can be represented as a knowledge graph where nodes represent customers and products and where typed edges represent customer reviews and purchasing events. In the medical domain, there are several knowledge graphs that model diseases, symptoms, drugs, genes, and their interactions (cf. [Ashburner et al., 2000, Wishart et al., 2008]). Increasingly, entities in these knowledge graphs are associated with visual data. For instance, in the online retail domain, there are product and advertising images and in the medical domain, there are patient-associated imaging data sets (MRIs, CTs, and so on). In addition, visual data is a large part of social networks and, in general, the world wide web.

Knowledge graphs facilitate the integration, organization, and retrieval of structured data and support various forms of search applications. In recent years KGs have been playing an increasingly crucial role in fields such as question answering [Das et al., 2017],

---

[1]Project URL: https://github.com/nle-ml/mmkb.git.

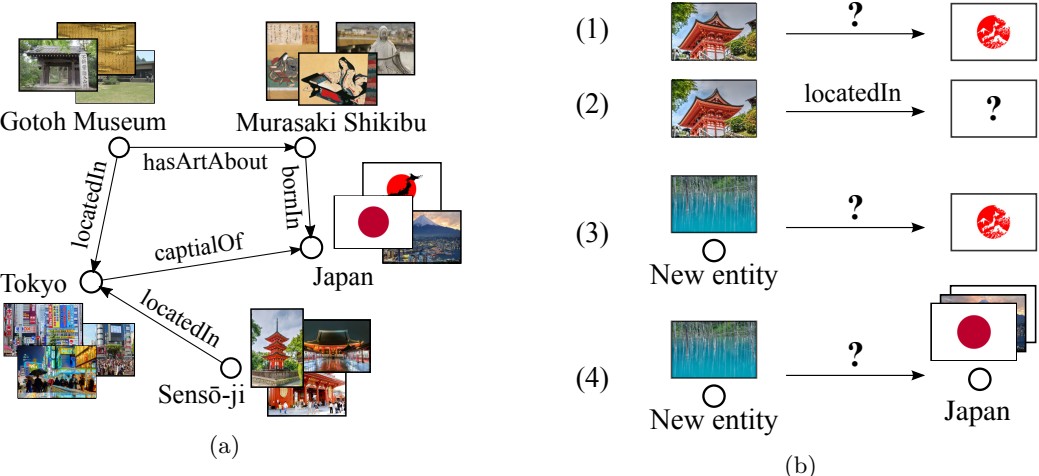

Figure 1: (a) a small part of a visual-relational knowledge graph and a set of query types; and (b) some visual-relational query types;

language modeling [Ahn et al., 2016], and text generation [Serban et al., 2016]. Even though there is a large body of work on constructing and maintaining KGs, the setting of visual-relational KGs, where entities are associated with visual data, has not received much attention. A visual-relational KG represents entities, relations between these entities, and a large number of images associated with the entities (see Figure 1a for an example). While IMAGENET [Deng et al., 2009] and the VISUALGENOME [Krishna et al., 2016] datasets are based on KGs such as WordNet they are predominantly used as either an object classification data set as in the case of IMAGENET or to facilitate scene understanding in a single image. With this work, we address the problem of reasoning about visual concepts across a large set of images organized in a knowledge graph. We want to explore to what extent web-extracted visual data can be used to enrich existing KGs so as to facilitate complex visual search applications going beyond basic image retrieval.

The core idea of our work is to treat images as first-class citizens both in KGs and visual-relational queries. The main objective of our work is to understand to what extent visual data associated with entities of a KG can be used in conjunction with deep learning methods to answer these visual-relational queries. Allowing images to be arguments of queries facilitates numerous novel query types. In Figure 1b we list some of the query types we address in this paper. In order to answer these queries, we built on KG embedding methods as well as deep representation learning approaches for visual data. This allows us to answer these visual queries both accurately and efficiently.

There are numerous application domains that could benefit from query answering in visual KGs. For instance, in online retail, visual representations of novel products could be leveraged for zero-shot product recommendations. Crucially, instead of only being able to retrieve similar products, a visual-relational KG would support the prediction of product attributes and more specifically what attributes customers might be interested in. For instance, in the fashion industry visual attributes are crucial for product recommendations [Liu et al.,

Table 1: Statistics of the knowledge graphs used in this paper.

| | Entities | Relations | Triples | | | Images | | |
|---|---|---|---|---|---|---|---|---|
| | $\mathcal{E}$ | $\mathcal{R}$ | Train | Valid | Test | Train | Valid | Test |
| ImageNet [Deng et al., 2009] | 21,841 | 18 | - | | | 14,197,122 | | |
| VisualGenome [Krishna et al., 2016] | 75,729 | 40,480 | 1,531,448 | | | 108,077 | | |
| FB15k [Bordes et al., 2013] | 14,951 | 1,345 | 483,142 | 50,000 | 59,071 | 0 | 0 | 0 |
| ImageGraph | 14,870 | 1,330 | 460,406 | 47,533 | 56,071 | 411,306 | 201,832 | 216,793 |

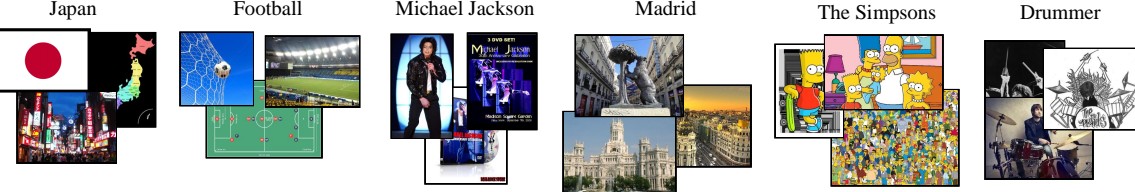

Figure 2: Image samples for some entities of ImageGraph.

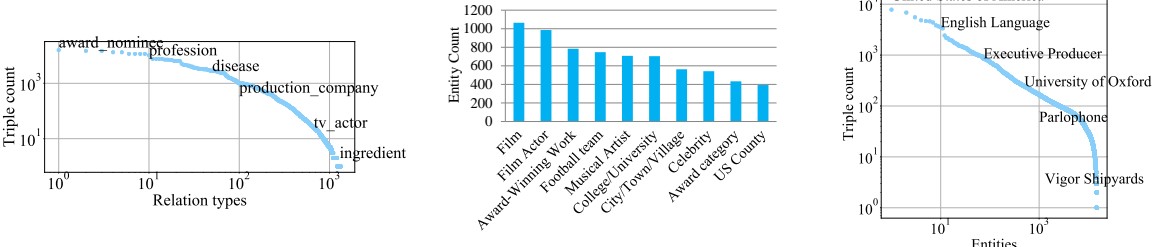

Figure 3: (Left) The distribution of relation types; (center) the 10 most frequent entity types; and (right) the distribution of entities in ImageGraph.

2016, Veit et al., 2015]. Being able to ground novel visual concepts into an existing KG with attributes and various relation types is a reasonable approach to zero-shot learning.

We make the following contributions. First, we introduce ImageGraph, a visual-relational web-extracted KG with 1,330 relations where 829,931 images are associated with 14,870 different entities. Second, we introduce a new set of visual-relational query types. Third, we propose a novel set of neural architectures and objectives that we use for answering these novel query types. These query types generalize image retrieval and link prediction queries. This is the first time that deep CNNs and KG embedding learning objectives are combined into a joint model. Fourth, we show that the proposed class of deep neural networks are also successful for zero-shot learning, that is, creating relations between entirely unseen entities and the KG using only visual data at query time.

## 2. Related Work

We discuss the relation of our contributions to previous work with an emphasis on relational learning, image retrieval, object detection, scene understanding, existing data sets, and zero-shot learning.

Relational Learning

There has been a flurry of approaches tailored to specific problems such as link prediction in multi-relational graphs. Examples are knowledge base factorization and embedding approaches [Bordes et al., 2013, Nickel et al., 2011, Guu et al., 2015] and random-walk based ML models [Lao et al., 2011, Gardner and Mitchell, 2015]. More recently, the focus has been on integrating additional attribute types such as text [Yahya et al., 2016, C. et al., 2017], temporal graph dynamics [Trivedi et al., 2017], and multiple modalities [Pezeshkpour et al., 2018]. Another line of research is concerned with extensions of the link prediction problem to multi-hop reasoning [Zhang et al., 2018]. We cannot list all prior link prediction methods here and instead refer the reader to two survey papers [Nickel et al., 2016a, Al Hasan and Zaki, 2011]. Contrary to existing approaches, we address the problem of answering visual-relational queries in knowledge graphs where the entities are associated with web-extracted images. We also address the zero-shot learning scenario, a problem that has not been addressed in the context of link prediction in multi-relational graphs.

Image ranking

Image retrieval is a popular problem and has been addressed by several authors [Wang et al., 2014, Yang et al., 2016, Jiang et al., 2017, Niu et al., 2018, Guy et al., 2018]. In [Yang et al., 2016] a re-ranking of the output of a given search engine by learning a click-based multi-feature similarity is proposed. The authors performed spectral clustering and obtained the final ranked results by computing click-based clusters. In [Guy et al., 2018] the authors fine-tune a DNN to rank photos a user might like to share in social media as well as a mechanism to detect duplicates. In [Niu et al., 2018] a joint user-image embedding is learned to generate a ranking based on user preferences. Contrary to these previous approaches we introduce a set of novel visual query types in a web-extracted KG with images and provide methods to answer these queries efficiently.

Relational and Visual Data

Previous work on combining relational and visual data has focused on object detection [Felzenszwalb et al., 2010, Girshick et al., 2014, Russakovsky et al., 2013, Marino et al., 2017, Li et al., 2017] and scene recognition [Doersch et al., 2013, Pandey and Lazebnik, 2011, Sadeghi and Tappen, 2012, Xiao et al., 2010, Teney et al., 2017] which are required for more complex visual-relational reasoning. Recent years have witnessed a surge in reasoning about human-object, object-object, and object-attribute relationships [Gupta et al., 2009, Farhadi et al., 2009, Malisiewicz and Efros, 2009, Yao and Fei-Fei, 2010, Felzenszwalb et al., 2010, Chen et al., 2013, Izadinia et al., 2014, Zhu et al., 2014]. The VisualGenome project [Krishna et al., 2016] is a knowledge base that integrates language and vision modalities. The project provides a knowledge graph, based on WordNet, which provides annotations of categories, attributes, and relation types for each image. Recent work has used the dataset to focus on scene understanding in single images. For instance, Lu *et al.* [Lu et al., 2016] proposed a model to detect relation types between objects depicted in an image by inferring sentences such as "man riding bicycle." Veit *et al.* [Veit et al., 2015] propose a siamese CNN to learn a metric representation on pairs of textile products so as to learn which products have similar styles. There is a large body of work on metric learning where the objective is to generate

image embeddings such that a pairwise distance-based loss is minimized [Schroff et al., 2015, Bell and Bala, 2015, Oh Song et al., 2016, Sohn, 2016, Wang et al., 2017]. Recent work has extended this idea to directly optimize a clustering quality metric [Song et al., 2017]. In Vincent *et al.* [Vincent et al., 2017] they proposed a mutual embedding space for images and knowledge graphs so the relationships between an image and known entities in a knowledge graph are jointly encoded. Zhou *et al.* [Zhou and Lin, 2016] propose a method based on a bipartite graph that links depictions of meals to its ingredients. Johnson *et al.* [Johnson et al., 2015] propose to use the VisualGenome data to recover images from text queries. In the work of Thoma *et al.* [Thoma et al., 2017], they merge in a joint representation the embeddings from images, text, and KG and use the representation to perform link prediction on DBpedia [Lehmann et al., 2015]. ImageGraph is different from these data sets in that the relation types hold between different images and image annotated entities. This defines a novel class of problems where one seeks to answer queries such as "*How are these two images related?*" With this work, we address problems ranging from predicting the relation types for image pairs to multi-relational image retrieval.

### Zero-shot Learning

We focus on exploring ways in which KGs can be used to find relationships between visual data of unseen entities, that is, entities not part of the KG during training, and visual data of known KG entities. This is a form of zero-shot learning (ZSL) where the objective is to generalize to novel visual concepts. Generally, ZSL methods (*e.g.* [Romera-Paredes and Torr, 2015, Zhang and Saligrama, 2015]) rely on an underlying embedding space, such as one based on visual attributes, to recognize unseen categories. With this paper, we do not assume the availability of such a common embedding space but we assume the existence of an external visual-relational KG. Similar to our approach, when this explicit knowledge is not encoded in the underlying embedding space, other works rely on finding the similarities through linguistic patterns (*e.g.* [Ba et al., 2015, Lu et al., 2016]), leveraging distributional word representations so as to capture a notion of similarity. These approaches, however, address scene understanding in a single image, *i.e.* these models are able to detect the visual relationships in one given image. Our approach, on the other hand, finds relationships between different images and entities.

## 3. ImageGraph: A Web-Extracted Visual Knowledge Graph

ImageGraph is a visual-relational KG whose relational structure is based on Freebase [Bollacker et al., 2008] and, more specifically, on FB15k, a subset of FreeBase and a popular benchmark data set [Nickel et al., 2016a]. Since FB15k does not include visual data, we perform the following steps to enrich the KG entities with image data. We implemented a web crawler that is able to parse query results for the image search engines Google Images, Bing Images, and Yahoo Image Search. To minimize the amount of noise due to polysemous entity labels (for example, there are more than 100 Freebase entities with the text label "Springfield") we extracted, for each entity in FB15k, all Wikipedia URIs from the 1.9 billion triple Freebase RDF dump. For instance, for Springfield, Massachusetts, we obtained such URIs as `Springfield_(Massachusetts,United_States)` and `Springfield_(MA)`. These URIs were processed and used as search queries for disambiguation purposes. We used

| Relation type | Example $(h, r, t)$ |
|---|---|
| Symmetric | (EmmaThompson,sibling,SophieThompson)
(SophieThompson,sibling,EmmaThompson) |
| Asymmetric | (Non-profitorganization,company_type,ApacheSoftwareFoundation)
(Statistics,students_majoring,PhD) |
| Others | (StarWars,film_series,StarWars)
(StarWarsEpisodeI:ThePhantomMenace,film_series,StarWars)
(StarWarsEpisodeII:AttackoftheClones,film_series,StarWars) |

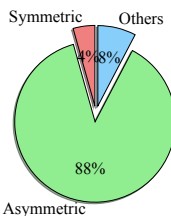

Figure 4: (Left) Example triples for *symmetric*, *asymmetric* and *others* relation types. (Right) Fraction of symmetric, asymmetric, and other relation types among all relation types in IMAGEGRAPH.

the crawler to download more than 2.4M images (more than 462Gb of data). We removed corrupted, low quality, and duplicate images and we used the 25 top images returned by each of the image search engines whenever there were more than 25 results. The images were scaled to have a maximum height or width of 500 pixels while maintaining their aspect ratio. This resulted in 829,931 images associated with 14,870 different entities (55.8 images per entity). After filtering out triples where either the head or tail entity could not be associated with an image, the visual KG consists of 564,010 triples expressing 1,330 different relation types between 14,870 entities. We provide three sets of triples for training, validation, and testing plus three more image splits also for training, validation and test. Table 1 lists the statistics of the resulting visual KG. Any KG derived from FB15K such as FB15K-237[Toutanova and Chen, 2015] can also be associated with the crawled images. Since providing the images themselves would violate copyright law, we provide the code for the distributed crawler and the list of image URLs crawled for the experiments in this paper[2].

The distribution of relation types is depicted in Figure 3 (left). It plots for each relation type the number of triples it occurs in. Some relation types such as award_nominee or profession occur quite frequently while others such as ingredient have only few instances. 4% of the relation types are symmetric, 88% are asymmetric, and 8% are others (see Table 4 (left)). Table 4 (right) lists specific instances of some relation types. There are 585 distinct entity types such as Person, Athlete, and City. Figure 3 (center) shows the most frequent entity types. Figure 3 (right) visualizes the distribution of entities in the triples of IMAGEGRAPH and some example entities.

Table 1 lists some statistics of the IMAGEGRAPH KG and other KGs from related work. First, we would like to emphasize the differences between IMAGEGRAPH and the Visual Genome project (VG) [Krishna et al., 2016]. With IMAGEGRAPH we address the problem of learning a representation for a KG with canonical relation types and not for relation types expressed through text. On a high level, we focus on answering visual-relational queries in a web-extracted KG. This is related to information retrieval except that in our proposed work, images are first-class citizens and we introduce novel and more complex query types. In contrast, VGD is focused on modeling relations between objects in images and the relation types are expressed in natural language. Additional differences between IMAGEGRAPH and IMAGENET are the following. IMAGENET is based on WORDNET a lexical

---

[2]IMAGEGRAPH crawler and URLs: https://github.com/robegs/imageDownloader.

database where synonymous words from the same lexical category are grouped into synsets. There are 18 relations expressing connections between synsets. In FREEBASE, on the other hand, there are two orders of magnitudes more relations. In FB15K, the subset we focus on, there are 1,345 relations expressing location of places, positions of basketball players, and gender of entities. Moreover, entities in IMAGENET exclusively represent entity types such as `Cats` and `Cars` whereas entities in FB15K are either entity types or instances of entity types such as `Albert Einstein` and `Paris`. This renders the computer vision problems associated with IMAGEGRAPH more challenging than those for existing datasets. Moreover, with IMAGEGRAPH the focus is on learning relational ML models that incorporate visual data both during learning and at query time.

## 4. Representation Learning for Visual-Relational Graphs

A knowledge graph (KG) $\mathcal{K}$ is given by a set of triples $\mathbf{T}$, that is, statements of the form $(\mathtt{h}, \mathtt{r}, \mathtt{t})$, where $\mathtt{h}, \mathtt{t} \in \mathcal{E}$ are the head and tail entities, respectively, and $\mathtt{r} \in \mathcal{R}$ is a relation type. Figure 1a depicts a small fragment of a KG with relations between entities and images associated with the entities. Prior work has not included image data and has, therefore, focused on the following two types of queries. First, the query type $(\mathtt{h}, \mathtt{r}?, \mathtt{t})$ asks for the relations between a given pair of head and tail entities. Second, the query types $(\mathtt{h}, \mathtt{r}, \mathtt{t}?)$ and $(\mathtt{h}?, \mathtt{r}, \mathtt{t})$, asks for entities correctly completing the triple. The latter query type is often referred to as knowledge base completion. Here, we focus on queries that involve visual data as query objects, that is, objects that are either contained in the queries, the answers to the queries, or both.

### 4.1 Visual-Relational Query Answering

When entities are associated with image data, several completely novel query types are possible. Figure 1b lists the query types we focus on in this paper. We refer to images used during training as *seen* and all other images as *unseen*.

(1) Given a pair of *unseen* images for which we do *not* know their KG entities, determine the *unknown* relations between the underlying entities.

(2) Given an *unseen* image, for which we do *not* know the underlying KG entity, and a relation type, determine the *seen* images that complete the query.

(3) Given an *unseen* image of an *entirely new entity* that is not part of the KG, and an *unseen* image for which we do *not* know the underlying KG entity, determine the *unknown* relations between the two underlying entities.

(4) Given an *unseen* image of an *entirely new entity* that is not part of the KG, and a known KG entity, determine the *unknown* relations between the two entities.

For each of these query types, the sought-after relations between the underlying entities have never been observed during training. Query types (3) and (4) are a form of zero-shot learning since neither the new entity's relationships with other entities nor its images have been observed during training. These considerations illustrate the novel nature of the visual

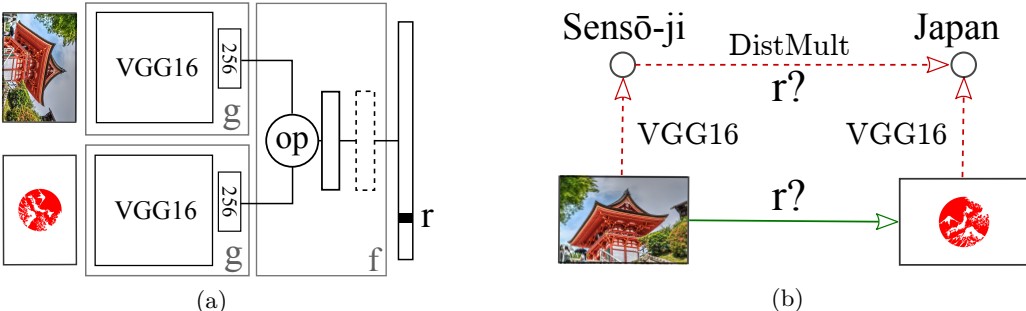

Figure 5: (a) the proposed architecture for query answering; and (b) an illustration of two possible approaches to visual-relational query answering. One can predict relation types between two images directly (green arrow; our approach) or combine an entity classifier with a KB embedding model for relation prediction (red arrows; baseline `VGG16+DistMult`).

query types. The machine learning models have to be able to learn the relational semantics of the KG and not simply a classifier that assigns images to entities. These query types are also motivated by the fact that for typical KGs the number of entities is orders of magnitude greater than the number of relations.

### 4.2 Deep Representation Learning for Visual-Relational Query Answering

We first discuss KG completion methods and translate the concepts to query answering in visual-relational KGs. Let $\mathbf{raw_i}$ be the raw feature representation for entity $\mathtt{i} \in \mathcal{E}$ and let $f$ and $g$ be differentiable functions. Most KG completion methods learn an embedding of the entities in a vector space via some *scoring function* that is trained to assign high scores to correct triples and low scores to incorrect triples. Scoring functions have often the form $f_\mathbf{r}(\mathbf{e_h}, \mathbf{e_t})$ where $\mathbf{r}$ is a relation type, $\mathbf{e_h}$ and $\mathbf{e_t}$ are $d$-dimensional vectors (the *embeddings* of the head and tail entities, respectively), and where $\mathbf{e_i} = g(\mathbf{raw_i})$ is an *embedding function* that maps the raw input representation of entities to the embedding space. In the case of KGs without visual data, the raw representation of an entity is simply its one-hot encoding.

Existing KG completion methods use the embedding function $g(\mathbf{raw_i}) = \mathbf{raw_i^\intercal W}$ where $\mathbf{W}$ is a $|\mathcal{E}| \times d$ matrix, and differ only in their scoring function, that is, in the way the embeddings of the head and tail entities are combined with the parameter vector $\boldsymbol{\phi_r}$:

- Difference (TRANSE[Bordes et al., 2013]): $f_\mathbf{r}(\mathbf{e_h}, \mathbf{e_t}) = -||\mathbf{e_h} + \boldsymbol{\phi_r} - \mathbf{e_t}||_2$ where $\boldsymbol{\phi_r}$ is a $d$-dimensional vector;

- Multiplication (DISTMULT[Yang et al., 2014]): $f_\mathbf{r}(\mathbf{e_h}, \mathbf{e_t}) = (\mathbf{e_h} * \mathbf{e_t}) \cdot \boldsymbol{\phi_r}$ where $*$ is the element-wise product and $\boldsymbol{\phi_r}$ a $d$-dimensional vector;

- Circular correlation (HOLE[Nickel et al., 2016b]): $f_\mathbf{r}(\mathbf{e_h}, \mathbf{e_t}) = (\mathbf{e_h} \star \mathbf{e_t}) \cdot \boldsymbol{\phi_r}$ where $[\mathbf{a} \star \mathbf{b}]_k = \sum_{i=0}^{d-1} \mathbf{a}_i \mathbf{b}_{(i+k) \bmod d}$ and $\boldsymbol{\phi_r}$ a $d$-dimensional vector; and

- Concatenation: $f_\mathbf{r}(\mathbf{e_h}, \mathbf{e_t}) = (\mathbf{e_h} \odot \mathbf{e_t}) \cdot \boldsymbol{\phi_r}$ where $\odot$ is the concatenation operator and $\boldsymbol{\phi_r}$ a $2d$-dimensional vector.

For each of these instances, the matrix $\mathbf{W}$ (storing the entity embeddings) and the vectors $\boldsymbol{\phi}_{\mathbf{r}}$ are learned during training. In general, the parameters are trained such that $f_{\mathbf{r}}(\mathbf{e_h}, \mathbf{e_t})$ is high for true triples and low for triples assumed not to hold in the KG. The training objective is often based on the logistic loss, which has been shown to be superior for most of the composition functions [Trouillon et al., 2016],

$$\min_{\Theta} \sum_{(\mathtt{h},\mathtt{r},\mathtt{t}) \in \mathbf{T_{pos}}} \log(1 + \exp(-f_{\mathbf{r}}(\mathbf{e_h}, \mathbf{e_t}))) + \sum_{(\mathtt{h},\mathtt{r},\mathtt{t}) \in \mathbf{T_{neg}}} \log(1 + \exp(f_{\mathbf{r}}(\mathbf{e_h}, \mathbf{e_t}))) + \lambda ||\Theta||_2^2, \quad (1)$$

where $\mathbf{T_{pos}}$ and $\mathbf{T_{neg}}$ are the set of positive and negative training triples, respectively, $\Theta$ are the parameters trained during learning and $\lambda$ is a regularization hyperparameter. For the above objective, a process for creating corrupted triples $\mathbf{T_{neg}}$ is required. This often involves sampling a random entity for either the head or tail entity. To answer queries of the types $(\mathtt{h}, \mathtt{r}, \mathtt{t}?)$ and $(\mathtt{h}?, \mathtt{r}, \mathtt{t})$ after training, we form all possible completions of the queries and compute a ranking based on the scores assigned by the trained model to these completions.

For the queries of type $(\mathtt{h}, \mathtt{r}?, \mathtt{t})$ one typically uses the softmax activation in conjunction with the categorical cross-entropy loss, which does not require negative triples

$$\min_{\Theta} \sum_{(\mathtt{h},\mathtt{r},\mathtt{t}) \in \mathbf{T_{pos}}} -\log\left(\frac{\exp(f_{\mathbf{r}}(\mathbf{e_h}, \mathbf{e_t}))}{\sum_{\mathbf{r} \in \mathcal{R}} \exp(f_{\mathbf{r}}(\mathbf{e_h}, \mathbf{e_t}))}\right) + \lambda ||\Theta||_2^2, \quad (2)$$

where $\Theta$ are the parameters trained during learning.

For visual-relational KGs, the input consists of raw image data instead of the one-hot encodings of entities. The approach we propose builds on the ideas and methods developed for KG completion. Instead of having a simple embedding function $\mathtt{g}$ that multiplies the input with a weight matrix, however, we use deep convolutional neural networks to extract meaningful visual features from the input images. For the composition function $\mathtt{f}$ we evaluate the four operations that were used in the KG completion literature: difference, multiplication, concatenation, and circular correlation. Figure 5a depicts the basic architecture we trained for query answering. The weights of the parts of the neural network responsible for embedding the raw image input, denoted by $\mathtt{g}$, are tied. We also experimented with additional hidden layers indicated by the dashed dense layer. The composition operation $\mathbf{op}$ is either difference, multiplication, concatenation, or circular correlation. To the best of our knowledge, this is the first time that KG embedding learning and deep CNNs have been combined for visual-relationsl query answering.

## 5. Experiments

We conduct a series of experiments to evaluate the proposed approach. First, we describe the experimental set-up that applies to all experiments. Second, we report and interpret results for the different types of visual-relational queries.

### 5.1 General Set-up

We used CAFFE, a deep learning framework [Jia et al., 2014] for designing, training, and evaluating the proposed models. The embedding function $\mathtt{g}$ is based on the VGG16 model introduced in [Simonyan and Zisserman, 2014]. We pre-trained the VGG16 on the ILSVRC2012

data set derived from IMAGENET [Deng et al., 2009] and removed the softmax layer of the original VGG16. We added a 256-dimensional layer after the last dense layer of the VGG16. The output of this layer serves as the embedding of the input images. The reason for reducing the embedding dimensionality from 4096 to 256 is motivated by the objective to obtain an efficient and compact latent representation that is feasible for KGs with billion of entities. For the composition function f, we performed either of the four operations difference, multiplication, concatenation, and circular correlation. We also experimented with an additional hidden layer with ReLu activation. Figure 5a depicts the generic network architecture. The output layer of the architecture has a softmax or sigmoid activation with cross-entropy loss. We initialized the weights of the newly added layers with the Xavier method [Glorot and Bengio, 2010].

We used a batch size of 45 which was the maximal possible fitting into GPU memory. To create the training batches, we sample a random triple uniformly at random from the training triples. For the given triple, we randomly sample one image for the head and one for the tail from the set of training images. We applied SGD with a learning rate of $10^{-5}$ for the parameters of the VGG16 and a learning rate of $10^{-3}$ for the remaining parameters. It is crucial to use two different learning rates since the large gradients in the newly added layers would lead to unreasonable changes in the pretrained part of the network. We set the weight decay to $5 \times 10^{-4}$. We reduced the learning rate by a factor of 0.1 every 40,000 iterations. Each of the models was trained for 100,000 iterations.

Since the answers to all query types are either rankings of images or rankings of relations, we utilize metrics measuring the quality of rankings. In particular, we report results for hits@1 (hits@10, hits@100) measuring the percentage of times the correct relation was ranked highest (ranked in the top 10, top 100). We also compute the median of the ranks of the correct entities or relations and the Mean Reciprocal Rank (MRR) for entity and relation rankings, respectively, defined as follows:

$$\text{MRR} = \frac{1}{2|\mathbf{T}|} \sum_{(\mathtt{h},\mathtt{r},\mathtt{t}) \in \mathbf{T}} \left( \frac{1}{\mathtt{rank_{img(h)}}} + \frac{1}{\mathtt{rank_{img(t)}}} \right) \tag{3}$$

$$\text{MRR} = \frac{1}{|\mathbf{T}|} \sum_{(\mathtt{h},\mathtt{r},\mathtt{t}) \in \mathbf{T}} \frac{1}{\mathtt{rank_r}}, \tag{4}$$

where $\mathbf{T}$ is the set of all test triples, $\mathtt{rank_r}$ is the rank of the correct relation, and $\mathtt{rank_{img(h)}}$ is the rank of the highest ranked image of entity $\mathtt{h}$. For each query, we remove all triples that are also correct answers to the query from the ranking. All experiments were run on commodity hardware with 128GB RAM, a single 2.8 GHz CPU, and a NVIDIA 1080 Ti.

### 5.2 Visual Relation Prediction

Given a pair of unseen images we want to determine the relations between their underlying unknown entities. This can be expressed with $(\mathtt{img_h}, \mathtt{r}?, \mathtt{img_t})$. Figure 1b illustrates this query type which we refer to as visual relation prediction. We train the deep architectures using the training and validation triples and images, respectively. For each triple $(\mathtt{h}, \mathtt{r}, \mathtt{t})$ in the training data set, we sample one training image uniformly at random for both the head and the tail entity. We use the architecture depicted in Figure 5a with the softmax

Table 2: Results for the relation prediction problem.

| Model | Median | Hits@1 | Hits@10 | MRR |
|---|---|---|---|---|
| `VGG16+DistMult` | 94 | 6.0 | 11.4 | 0.087 |
| Prob. Baseline | 35 | 3.7 | 26.5 | 0.104 |
| DIFF | 11 | 21.1 | 50.0 | 0.307 |
| MULT | 8 | 15.5 | 54.3 | 0.282 |
| CAT | **6** | **26.7** | **61.0** | **0.378** |
| DIFF+1HL | 8 | 22.6 | 55.7 | 0.333 |
| MULT+1HL | 9 | 14.8 | 53.4 | 0.273 |
| CAT+1HL | **6** | 25.3 | 60.0 | 0.365 |

activation and the categorical cross-entropy loss. For each test triple, we sample one image uniformly at random from the test images of the head and tail entity, respectively. We then use the pair of images to query the trained deep neural networks. To get a more robust statistical estimate of the evaluation measures, we repeat the above process three times per test triple. Again, none of the test triples and images are seen during training nor are any of the training images used during testing. Computing the answer to one query takes the model 20 ms.

We compare the proposed architectures to two different baselines: one based on entity classification followed by a KB embedding method for relation prediction (`VGG16+DistMult`), and a probabilistic baseline (Prob. Baseline). The entity classification baseline consists of fine-tuning a pretrained VGG16 to classify images into the $14,870$ entities of ImageGraph. To obtain the relation type ranking at test time, we predict the entities for the head and the tail using the VGG16 and then use the KB embedding method DistMult[Yang et al., 2014] to return a ranking of relation types for the given (head, tail) pair. `DistMult` is a KB embedding method that achieves state of the art results for KB completion on FB15k [Kadlec et al., 2017]. Therefore, for this experiment we just substitute the original output layer of the VGG16 pretrained on ImageNet with a new output layer suitable for our problem. To train, we join the train an validation splits, we set the learning rate to $10^{-5}$ for all the layers and we train following the same strategy that we use in all of our experiments. Once the system is trained, we test the model by classifying the entities of the images in the test set. To train `DistMult`, we sample 500 negatives triples for each positive triple and used an embedding size of 100. Figure 5b illustrates the `VGG16+DistMult` baseline and contrasts it with our proposed approach. The second baseline (probabilistic baseline) computes the probability of each relation type using the set of training and validation triples. The baseline ranks relation types based on these prior probabilities.

Table 2 lists the results for the two baselines and the different proposed architectures. The probabilistic baseline outperforms the `VGG16+DistMult` baseline in 3 of the metrics. This is due to the highly skewed distribution of relation types in the training, validation, and test triples. A small number of relation types makes up a large fraction of triples. Figure 3 (left) and 3 (right) depicts the plots of the counts of relation types and entities. Moreover, despite DistMult achieving a hits@1 value of 0.46 for the relation prediction problem between entity pairs the baseline `VGG16+DistMult` performs poorly. This is due to the poor

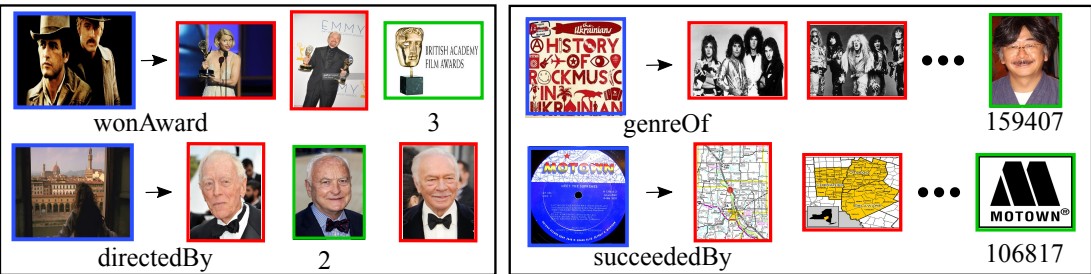

Figure 6: Example queries and qualitative results for the multi-relational image retrieval problem.

Table 3: Results for multi-relational image retrieval.

|  | Median | | Hits@100 | | MRR | |
|---|---|---|---|---|---|---|
| Model | Head | Tail | Head | Tail | Head | Tail |
| Baseline | 6504 | 2789 | 11.9 | 18.4 | **0.065** | **0.115** |
| DIFF | 1301 | 877 | 19.6 | 26.3 | 0.051 | 0.094 |
| MULT | 1676 | 1136 | 16.8 | 22.9 | 0.040 | 0.080 |
| CAT | 1022 | 727 | 21.4 | 27.5 | 0.050 | 0.087 |
| DIFF+1HL | 1644 | 1141 | 15.9 | 21.9 | 0.045 | 0.085 |
| MULT+1HL | 2004 | 1397 | 14.6 | 20.5 | 0.034 | 0.069 |
| CAT+1HL | 1323 | 919 | 17.8 | 23.6 | 0.042 | 0.080 |
| CAT-SIG | **814** | **540** | **23.2** | **30.1** | 0.049 | 0.082 |

entity classification performance of the VGG (accurracy: 0.082, F1: 0.068). In the remainder of the experiments, therefore, we only compare to the probabilistic baseline. In the lower part of Table 2, we lists the results of the experiments. DIFF, MULT, and CAT stand for the different possible composition operations. We omitted the composition operation circular correlation since we were not able to make the corresponding model converge, despite trying several different optimizers and hyperparameter settings. The post-fix 1HL stands for architectures where we added an additional hidden layer with ReLu activation before the softmax. The concatenation operation clearly outperforms the multiplication and difference operations. This is contrary to findings in the KG completion literature where MULT and DIFF outperformed the concatenation operation. The models with the additional hidden layer did not perform better than their shallower counterparts with the exception of the DIFF model. We hypothesize that this is due to difference being the only linear composition operation, benefiting from an additional non-linearity. Each of the proposed models outperforms the baselines.

## 5.3 Multi-Relational Image Retrieval

Given an unseen image, for which we do not know the underlying KG entity, and a relation type, we want to retrieve existing images that complete the query. If the image for the head

| | Median | | Hits@1 | | Hits@10 | | MRR | |
|---|---|---|---|---|---|---|---|---|
| | H | T | H | T | H | T | H | T |
| Zero-Shot Query (3) | | | | | | | | |
| Base | 34 | 31 | 1.9 | 2.3 | 18.2 | 28.7 | 0.074 | 0.089 |
| CAT | **8** | **7** | **19.1** | **22.4** | **54.2** | **57.9** | **0.306** | **0.342** |
| Zero-Shot Query (4) | | | | | | | | |
| Base | 9 | 5 | 13.0 | 22.6 | 52.3 | 64.8 | 0.251 | 0.359 |
| CAT | **5** | **3** | **26.9** | **33.7** | **62.5** | **70.4** | **0.388** | **0.461** |

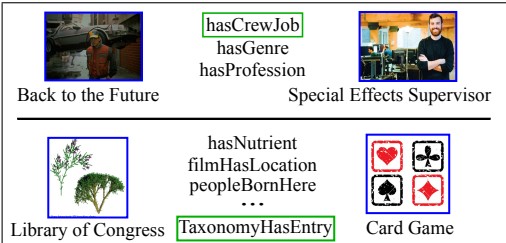

Figure 7: (Left) Results for the zero-shot learning experiments. (Right) Example results for zero-shot learning. For each pair of images the top three relation types (as ranked by the CAT model) are listed. For the pair of images at the top, the first relation type is correct. For the pair of images at the bottom, the correct relation type `TaxonomyHasEntry` is not among the top three relation types.

entity is given, we return a ranking of images for the tail entity; if the tail entity image is given we return a ranking of images for the head entity. This problem corresponds to query type (2) in Figure 1b. Note that this is equivalent to performing multi-relational metric learning which, to the best of our knowledge, has not been done before. We performed experiments with each of the three composition functions $f$ and for two different activation/loss functions. First, we used the models trained with the softmax activation and the categorical cross-entropy loss to rank images. Second, we took the models trained with the softmax activation and substituted the softmax activation with a sigmoid activation and the corresponding binary cross-entropy loss. For each training triple $(h, r, t)$ we then created two negative triples by sampling once the head and once the tail entity from the set of entities. The negative triples are then used in conjunction with the binary cross-entropy loss of equation 1 to refine the pretrained weights. Directly training a model with the binary cross-entropy loss was not possible since the model did not converge properly. Pretraining with softmax and categorical cross-entropy loss was crucial to make the binary loss work.

During testing, we used the test triples and ranked the images based on the probabilities returned by the respective models. For instance, given the query $(\text{img}_{\text{Sensō-ji}}, \text{locatedIn}, \text{img}_t?)$, we substituted $\text{img}_t?$ with all training and validation images, one at a time, and ranked the images according to the probabilities returned by the models. We use the rank of the highest ranked image belonging to the true entity (here: `Japan`) to compute the values for the evaluation measures. We repeat the same experiment three times (each time randomly sampling the images) and report average values. Again, we compare the results for the different architectures with a probabilistic baseline. For the baseline, however, we compute a distribution of head and tail entities for each of the relation types. For example, for the relation type `locatedIn` we compute two distributions, one for head and one for tail entities. We used the same measures as in the previous experiment to evaluate the returned image rankings.

Table 3 lists the results of the experiments. As for relation prediction, the best performing models are based on the concatenation operation, followed by the difference and multiplication operations. The architectures with an additional hidden layer do not improve the performance. We also provide the results for the concatenation-based model with softmax activation where

we refined the weights using a sigmoid activation and negative sampling as described before. This model is the best performing model. All neural network models are significantly better than the baseline with respect to the median and hits@100. However, the baseline has slightly superior results for the MRR. This is due to the skewed distribution of entities and relations in the KG (see Figure 3 (right) and Figure 3 (left)). This shows once more that the baseline is highly competitive for the given KG. Figure 6 visualizes the answers the CAT-SIG model provided for a set of four example queries. For the two queries on the left, the model performed well and ranked the correct entity in the top 3 (green frame). The examples on the right illustrate queries for which the model returned an inaccurate ranking. To perform query answering in a highly efficient manner, we precomputed and stored all image embeddings once, and only compute the scoring function (involving the composition operation and a dot product with $\phi_{\mathbf{r}}$) at query time. Answering one multi-relational image retrieval query (which would otherwise require 613,138 individual queries, one per possible image) took only 90 ms.

### 5.4 Zero-Shot Visual Relation Prediction

The last set of experiments addresses the problem of zero-shot learning. For both query types, we are given an new image of an entirely new entity that is not part of the KG. The first query type asks for relations between the given image and an unseen image for which we do not know the underlying KG entity. The second query type asks for the relations between the given image and an existing KG entity. We believe that creating multi-relational links to existing KG entities is a reasonable approach to zero-shot learning since the relations to existing visual concepts and their attributes provide a characterization of the new entity/category.

For the zero-shot experiments, we generated a new set of training, validation, and test triples. We randomly sampled 500 entities that occur as head (tail) in the set of test triples. We then removed all training and validation triples whose head or tail is one of these 1000 entities. Finally, we only kept those test triples with one of the 1000 entities either as head or tail but not both. For query type (4) where we know the target entity, we sample 10 of its images and use the models 10 times to compute a probability. We use the average probabilities to rank the relations. For query type (3) we only use one image sampled randomly. As with previous experiments, we repeated procedure three times and averaged the results. For the baseline, we compute the probabilities of relation in the training and validation set (for query type (3)) and the probabilities of relations conditioned on the target entity (for query type (4)). Again, these are very competitive baselines due to the skewed distribution of relations and entities. Table 7 (left) lists the results of the experiments. The model based on the concatenation operation (CAT) outperforms the baseline and performs surprisingly well. The deep models are able to generalize to unseen images since their performance is comparable to the performance in the relation prediction task (query type (1)) where the entity was part of the KG during training (see Table 2). Figure 7 (right) depicts example queries for the zero-shot query type (3). For the first query example, the CAT model ranked the correct relation type first (indicated by the green bounding box). The second example is more challenging and the correct relation type was not part of the top 10 ranked relation types. Figure 5.4 shows one concrete example of the zero-shot learning

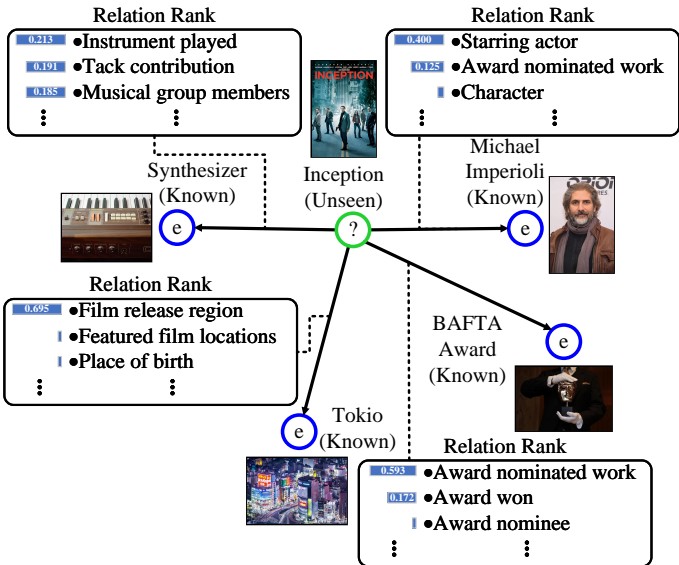

Figure 8: Qualitative example of the zero-shot learning problem. The plot shows the most probable relations that link a sample from an unknown entity (green) with samples of known entities (blue) of the KG.

problem. In green, visual data from an unknown entity is linked with visual data from KG entities (blue) by ranking the most probable relation types. This problem cannot be addressed with standard relation prediction methods since entities need to be part of the KG during training for these models to work.

## 6. Conclusion

KGs are at the core of numerous AI applications. Research has focused either on link prediction working only on the relational structure or on scene understanding in a single image. We present a novel visual-relational KG where the entities are enriched with visual data. We proposed several novel query types and introduce neural architectures suitable for probabilistic query answering. We propose a novel approach to zero-shot learning as the problem of visually mapping an image of an entirely new entity to a KG.

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
