# OpenReview forum: "Answering Visual-Relational Queries in Web-Extracted Knowledge Graphs"
_AKBC.ws/2019/Conference — AKBC 2019_

### Official Review · AnonReviewer2 · 2019-01-04

**Rating:** 4
**Confidence:** 4

**Review:**

The paper proposes to extend knowledge base completion benchmarks with visual data to explore novel query types that allow searching and completing knowledge bases by images. Experiments were conducted on standard KB completion tasks using images as entity representations instead of one-hot vectors, as well as a zero shot tasks using unseen images of unknown entities.

Overall I think that enriching KBs with visual data is appealing and important. Using images to query knowledge bases can be a practical tool for several applications. However, the overall experimental setup suffers from several problems. The results are overall very low. In the non-zero shot experiments I would like to see a comparison to using entity embeddings, and maybe even using a combination of both, as this is the more interesting setup. For instance, I would like to see whether using images as additional information can help building better entity representations. The explored link prediction models are all known, so apart from using images instead of entities there is very limited novelty. The authors find that concatenation followed by dot-product with relation-vector works best. This is very unfortunate because it means that there is no interaction between h and t at all, i.e.: s(h, r, t) = [h;t] * r = h * r_1 + t * r_2. This means that finding t given h,r only depends on r and not on h at all. Finally, this shows that the proposed image embeddings derived from the pretrained VGG16 model are not very useful for establishing relations.

Given the mentioned problems I can unfortunately not recommend this paper for acceptence.

Other comments:
- I wouldn't consider a combination of pretrained image embeddings bsaed on CNNs with KB embeddings a "novel machine learning approach", but rather a standard technique
- redefine operators when describing LP models: \odot is typically used for element-wise multiplication, for concatenation use  [h; t] for instance
- (head, relation, tail) is quite unusual -->  better: (subject, predicate, object)
- baselines are super weak. Concatenation should be the baseline as it connects h and t with r indepedent of each other. What is the probabilistic baseline?

---

> ### Author Response · Authors · 2019-01-22
> **Response to comments**
>
> Thank you for the helpful review.
>
> The novelty of our submission is in the visual KB that we have created, the novel query types we propose, and the combination of standard CNNs with KB embedding scoring functions. We do not claim that the scoring functions are novel. What is novel is the combination of CNNs on image data with these scoring functions for the novel query types we propose. Our work shows that some existing scoring functions are not better than a concatenation, at least for the image retrieval task. We have explored the zero-shot learning setting in the link prediction problem for the first time in the literature. We think that this also makes the work novel.
>
> Experiments show that the difficulty and performance are different from that of the traditional link prediction problem. We have (adapted and) benchmarked some of the most standard scoring functions from the link prediction literature in this problem and show that: i) they work, to some extent, in the relation prediction task, and ii) they perform poorly in the image retrieval task, where the (naive) concatenation approach performs the best. We remark again the complexity of the latter problem (see rebuttal for reviewer 1). We think that this set of experiments constitutes a solid answer to the question regarding the ability of the traditional scoring function to answer these new types of visual queries.
>
> Please also note the setup descriptions in Section 4.1.  In the image retrieval and link prediction scenarios, (experiments  (1) and (2)) we have images for which we do not know the underlying KG entities. This is the reason why we have not considered approaches that leverage entity embeddings, as this information is not available in our setting.
>
> Thank you for your suggestion regarding the notation used to define certain operators. We completely agree. However, we think the notation (head, relation, tail) is not that uncommon, as it has been used in previous works (e.g. (Bordes et al. 2013), (Garcia-Duran&Niepert, 2017)).
>
> Finally, the probabilistic baseline is explained in Section 5.2 at the end of the second paragraph: “The second baseline (probabilistic baseline) computes the
> probability of each relation type using the set of training and validation triples. The baseline
> ranks relation types based on these prior probabilities.”

---

> > ### Comment · AnonReviewer2 · 2019-01-28
> > **RE**
> >
> > Well, the biggest issues still remain though. Considering everything this work added images from a web search to an existing KG and showed that traditional approaches do not work well when using **a pretrained VGG16 model**, in fact they do not work at all. Don't get me wrong, the idea of exploring visual information for AKBC and using it for zero-shot learning is very interesting but the experiments are just too limited. End-to-end training should be considered instead of taking a pretrained feature extractor that doesn't work for this task, or at the very least other models for visual feature extraction should have been tried. The problem is that in its current form the experiments tell us only that the pretrained VGG16 model is a bad feature extractor but nothing conclusive about existing models for link prediction. Because of that unfortunately also the zero shot experiments add little value.

---

> > > ### Author Response · Authors · 2019-02-01
> > > **Response to comments**
> > >
> > > We respectfully disagree with the statement “the experiments tell us only that the pretrained VGG16 model is a bad feature extractor,” and we are not sure how you come to that conclusion. The results for the image retrieval tasks are mixed not because the VGG16 network is a bad feature extractor but because the entities and corresponding images are heterogeneous. For instance, there are entities such as “Albert Einstein” for which the feature extraction works very well, and other such as “USA” for which it is challenging because there is no canonical image representing said entity. We envision in the future models that pay attention to particular more canonical images in this case. We do not anticipate significant improvement by simply using a different feature extractor. Replacing the VGG16 network with a deeper or more advanced CNN, for example, should lead to minor improvements only.
> > > Moreover, we would like to clarify that the model is pre-trained on ImageNet, but the entire model is end-to-end fine tuned, with each of the scoring functions.
> > >
> > > The results for relation prediction and zero-shot learning are actually promising. Since the input for the system is an image, there is no information about the entity that it represents. Hence, the problem and the performance cannot be compared with previous methods as we present the problem (and possible solutions) for the first time.

---

> > > > ### Comment · AnonReviewer2 · 2019-02-01
> > > > **RE**
> > > >
> > > > I am sorry I misunderstood that the model was finetuned, but from the paper it was not obvious to me. Anyway my main issue remains,  the concatenation model doesn't compute entity interaction which leaves me with the conclusion that VGG is not a good feature extractor. So if the concatenation model works the way I understand it, it means that results cannot be promising because you are computing a score where entity representations do not interact at all which should be essential for any reasonable link prediction model.

---

### Official Review · AnonReviewer3 · 2019-01-05
**Compelling new task and dataset**

**Rating:** 9
**Confidence:** 4

**Review:**

The paper introduces several novel tasks for visual reasoning resembling knowledge base completion tasks but involving images linked to entities: finding relations between entities represented by images and finding images given an image and a relation. The task is accompanied with a new dataset, which links images crawled from the web to FreeBase entities. The authors propose and evaluate the first approach on this dataset.

The paper is well written and clearly positions the novelty of the contributions with respect to the related work.

Questions:
* What are the types of errors of the proposed approach? The error analysis is missing. A brief summary or a table based on the sample from the test set can provide insights of the limitations and future directions.
* Is this task feasible? In some cases information contained in the image can be insufficient to answer the query. Error analysis and human baseline would help to determine the expected upper-bound for this task.
* Which queries involving images and KG are not addressed in this work? The list of questions in 4.1. can be better structured, e.g. in a table/matrix: Target (relation/entity/image), Data (relation/entity/image)

---

> ### Author Response · Authors · 2019-01-22
> **Response to comments**
>
> We thank the reviewer for the encouraging review.
>
> We have observed that the proposed models learn to relate the two entity types that are involved in a certain relationship/predicate. This is illustrated in Figure 6. However, it sometimes struggles to find an image of the entity that completes the query. This is related to your concern regarding the feasibility of the task. This highly depends on the entities involved in the query. Queries involving entities with canonical images (“Statue of Liberty”) are easier to answer than those involving entities that can be represented with heterogeneous images (“United States of America”). This is a nice suggestion, so we will include this discussion in the paper.
>
> Our queries solely include visual information. This means that we do not address queries where the underlying entity for a given image is known.

---

### Official Review · AnonReviewer1 · 2019-01-09
**Interesting approach to a new task + new data set**

**Rating:** 7
**Confidence:** 4

**Review:**

This work aims to address the problem of answering visual-relational queries in knowledge graphs where the entities are associated with web-extracted images.

The paper introduces a newly constructed large scale visual-relational knowledge graph built by scraping the web. Going beyond previous data sets like VisualGenome having annotations within the image, the ImageGraph data set that this work proposes allows for queries over relations between multiple images and will be useful to the community for future work. Some additional details about the dataset would have been useful such as the criteria used to decide "low quality images" that were omitted from the web crawl as well as  the reason for omitting 15 relations and 81 entities from FB15k.

While existing relational-learning models on knowledge graphs employ an embedding matrix to learn a representation for the entity, this paper proposes to use deep neural networks to extract a representation for the images. By employing deep representations of images associated with previously unseen entities, their method is also able to answer questions by generalizing to novel visual concepts, providing the ability to zero-shot answer questions about these unseen entities.

The baselines reported by the paper are weak especially the VGG+DistMult baseline with very low classifier score leading to its uncompetitive result. It would be worth at this point to try and build a better classifier that allows for more reasonable comparison with the proposed method. (Accuracy 0.082 is really below par) As for the probabilistic baseline, it only serves to provide insights into the prior biases of the data and is also not a strong enough baseline to make the results convincing.

Well written paper covering relevant background work, but would be much stronger with better baselines.

---

> ### Author Response · Authors · 2019-01-22
> **Response to comments**
>
> Thank you for your helpful review.
>
> FB15k contain entities like “ISO_3166-1:SO”, or “C16:1(n-7)”, for which the crawler returned few images, or even no image at all. After removing those entities from the graph, some relations were not present in the data set.
>
> We could have replaced our feature extractor VGG-16 with more sophisticated/up-to-date CNNs such as ResNet or DenseNet. These models obtain moderate gains in accuracy on ImageNet, but they come at the cost of (largely) slower running times. In our opinion, the low performance relates to the difficulty of the problem. Our data set has 15 times more of categories/entities than ImageNet, hence the low performance in the image retrieval task. However, for the relation prediction task, where the output space is comparable to that of ImageNet, performances are much more competitive.
>
> We agree that the experiments provide insights into the bias of the data set. Most importantly, the experiments show that the introduced problem is much harder than the traditional link prediction problem. We wanted to evaluate the scoring functions of state-of-the-art link prediction methods on the problems. At least for the visual entity prediction problem, these scoring functions did not work better than a concatenation. We hope this serves as a first step to this new problem.

---

### Meta-Review · Area_Chair1 · 2019-02-11
**New useful dataset needs to make less strong claims about novelty**

**Recommendation:** Accept (Poster)
**Confidence:** 4

**Metareview:**

This paper introduces a useful new dataset called ImageGraph that allows for the assessment of tasks on the combination of images and knowledge graphs. The paper presents a number of tasks over that datasets and architectures to address those tasks. The reviewers agree that the baselines could be improved upon and there is a question as to whether the architectures are promising or not.

I think the paper should be accepted because the dataset is fundamentally useful and the authors establish good baselines for the considered tasks. Additionally, using KGs with images together is really promising. However, joint image + kg embeddings have already been investigated elsewhere see [1]. I would recommend that the authors soften their claims of novelty. The work is useful and points to a number of good directions for future work.

[1] Towards Holistic Concept Representations: Embedding Relational Knowledge, Visual Attributes, and Distributional Word Semantics. S Thoma, A Rettinger, F Both. International Semantic Web Conference, 694-710

---

### Decision · Program_Chairs · 2019-02-15
**AKBC 2019 Conference Decision**

Accept